# DIMPLE: DISCRETE DIFFUSION MULTIMODAL LARGE LANGUAGE MODEL WITH PARALLEL DECODING

## ABSTRACT

In this work, we present Dimple and Dimple+, two Discrete Diffusion Multimodal Large Language Models (dMLLMs). Dimple is initialized from a discrete diffusion Large Language Model (dLLM) without multimodal understanding ability, and learns such ability through a hybrid training paradigm that first applies autoregressive training and then switches to discrete diffusion training. Dimple+ is initialized from an autoregressive Multimodal Large Language Models, and acquires parallel decoding capability through pure discrete diffusion training. Both models achieve performance comparable to their autoregressive baselines, and Dimple+ establishes new state-of-the-art results among dMLLMs. To enhance inference efficiency, we propose Confident Decoding, which dynamically adjusts the number of tokens generated per iteration. Experiments show that it accelerates decoding by 2×–6× with only minor performance degradation. We also demonstrate that the Prefilling technique, previously used in autoregressive models, can be effectively applied to dMLLMs with bidirectional attention, achieving nearly lossless speedups of 1.7×–7×. Finally, we introduce the Structure Prior method, enabling fine-grained control over response format and reasoning structure, which is difficult to realize in autoregressive models.

## 1 INTRODUCTION

Recent months have witnessed a surge of interest in applying diffusion models to natural language processing tasks. These Discrete Diffusion Large Language Models (dLLMs) reframe generation as a denoising process, allowing parallel decoding and improved control over output structure. The first general framework for diffusion over discrete spaces was introduced in D3PM (Campbell et al., 2024), which set up the mathematical foundation of the diffusion langauge model. Subsequently, RDM (Zheng et al., 2024), MD4 (Shi et al., 2025b), and MDLM (Sahoo et al., 2024b) reparameterized and simplified the loss for discrete diffusion into a weighted cross-entropy form, which was widely adopted in later researches (Nie et al., 2025; Ye et al., 2025b; Gong et al., 2025; Zhu et al., 2025). Other explorations include extending discrete diffusion models to continuous time (Campbell et al., 2022), discrete flow matching (Gat et al., 2024) and concrete score matching (Meng et al., 2022; Sun et al., 2023). However, these works were typically conducted on models under 2B parameters. Only more recently, LLaDA (Nie et al., 2025), LLaDA 1.5 (Zhu et al., 2025), Diffu-GPT/LLaMA (Gong et al., 2025), and Dream (Ye et al., 2025b) successfully scaled up dLLMs to much larger sizes.

Concurrently, the field of multimodal large language models (MLLMs) has advanced rapidly, with models such as LLaVA (Li et al., 2024), Qwen-VL (Bai et al., 2025), and InternVL (Mohbat & Zaki, 2024) achieving state-of-the-art results on a range of vision-language benchmarks. However, most of the existing MLLMs rely exclusively on autoregressive generation mechanisms.

In this work, we introduce **Dimple** and **Dimple+** , two *Discrete Diffusion Multimodal Large Language Models* (dMLLMs). Due to the inefficiency of discrete diffusion training in learning multimodal understanding ability, Dimple and Dimple+ utilize a "first-autoregressive-then-diffusion" training strategy and "autoregressive model initialization" strategy, respectively. Dimple begins with Dream Ye et al. (2025b), a dLLM lacking multimodal understanding ability, and first undergoes autoregressive training followed by the diffusion training. Ablation studies in the supplementary material show that this hybrid approach is significantly more effective for learning multimodal understanding ability than the pure diffusion training. In contrast, Dimple+ is initialized from Qwen2.5-VL (Bai et al., 2025),

an MLLM with strong multimodal understanding ability but without discrete diffusion-based parallel decoding ability. By applying only discrete diffusion training, Dimple+ acquires the parallel decoding capability while inheriting Qwen2.5-VL's strong multimodal understanding ability, achieving the SOTA performance among dMLLMs.

To further improve inference efficiency and controllability, we propose **Confident Decoding**, which dynamically adjusts the number of tokens decoded per iteration based on a confidence threshold. This technique accelerates decoding by approximately 2× to 6× with only minor performance degradation. We also demonstrate the feasibility of applying the **Prefilling** technique in autoregressive models to dMLLMs with bidirectional attention. Prefilling achieves nearly lossless speedups of 1.7× to 7× on most benchmarks. In addition, we explore the **Structure Prior** technique, which enables fine-grained control over response structure. With Structure Prior, dMLLMs can perform early answer prediction, structured reasoning, and in-place output formatting, which are difficult to achieve with autoregressive models.

**Contributions.**   Our key contributions are as follows:

- We introduce **Dimple** and **Dimple+**, both of which achieve performance comparable to their autoregressive baselines under the same training budget. Moreover, Dimple+ establishes state-of-the-art results among dMLLMs.
- We propose **Confident Decoding**, re-implement **Prefilling**, and explore **Structure Prior** for dMLLM inference that improve the efficiency, flexibility, and controllability.

## 2 PRELIMINARY

**Discrete Diffusion Large Language Models (dLLMs).**   dLLMs are a class of generative models that conceptualize text generation as a denoising process over discrete time steps. Let $x_0 \sim p_{\text{data}}(x_0)$ denote the original token sequence, and $x_t$ its noisy version at time $t \in [0, T]$. The forward noising process is defined as a Markov chain $q(x_{1:T}|x_0) = \prod_{t=1}^{T} q(x_t|x_{t-1})$, progressively adding noise to $x_0$. For discrete token modeling with an absorbing state, this is formalized as:

$$q(x_t|x_{t-1}) = \text{Cat}(x_t; Q_t^\top x_{t-1}), \quad q(x_t|x_0) = \alpha_t x_0 + (1-\alpha_t)\mathbf{m}, \quad \alpha_t = \prod_{i=1}^{t}(1-\beta_i), \quad (1)$$

where $Q_t = (1-\beta_t)I + \beta_t \mathbf{1}\mathbf{m}^\top$, with $\mathbf{m}$ being the one-hot representation of a special [MASK] token. The reverse (generative) process is modeled as:

$$p_\theta(x_{0:T}) = p_\theta(x_T)\prod_{t=1}^{T} p_\theta(x_{t-1}|x_t) = \prod_{t=1}^{T} q(x_{t-1}|x_0)p_\theta(x_0|x_t), \quad (2)$$

where each transition $p_\theta(x_{t-1}|x_t)$ is learned to approximate the reverse of the noising process with bidirectional attention. The training loss of Discrete Diffusion, derived from the variational lower bound of $p(x_0)$, is a weighted cross-entropy loss:

$$\mathcal{L}_D = \mathbb{E}_t[\mathcal{L}_t], \quad \mathcal{L}_t = \mathbb{E}_{q(x_t|x_0)}\left[-\sum_{n=1}^{N}\frac{1}{t}\delta_{x_t^n,\mathbf{m}}(x_0^n)\,log[f_\theta(x_t)^n]\right]. \quad (3)$$

This loss can be implemented as a Masked Language Modeling loss with a random mask ratio. The mask ratio is function of the time index $t$, which is a random variable. The expectation $\mathbb{E}_t$ is taken over a uniform distribution of time steps $t \in [0, 1]$. In practice, for each training instance, one time index $t$ is samples to determine the proportion of tokens to be masked. The inner expectation $\mathbb{E}_{q(x_t|x_0)}$ is taken over the forward transition probability defined in Eq. 1. $N$ denotes the number of tokens in the current input sample $x_0$. Inside the summation $\sum_{n=1}^{N}$, a weighted cross-entropy is computed over each token. The weights consist of two parts: (1) a time-dependent factor $\frac{1}{t}$, and (2) an indicator function $\delta_{x_t^n,\mathbf{m}}$ which ensures that the loss is only computed for masked tokens. Finally, $f_\theta$ denotes the model's prediction.

**Challenges of Discrete Diffusion Training.** Due to the use of bidirectional attention and the limited number of sampled timesteps, training with Discrete Diffusion directly incurs higher computational costs compared to autoregressive (AR) training (Prabhudesai et al., 2025; Yu et al., 2025).

1. Compared to next-token prediction loss, masked language modeling has lower utilization of the training corpus. In Auto-Regressvie training, every response token receives supervision. In contrast, in discrete diffusion training, only tokens where $\delta_{x_t^n, \mathbf{m}} = 1$ (i.e., masked tokens) are used to compute the loss. Take linear masking scheduling as an example, with $t \sim \text{Uniform}(0, 1)$, only about half the tokens are supervised.

2. In diffusion training, the supervision signal does not cover the entire generation trajectory. Discrete diffusion inference proceeds iteratively as $x_T \rightarrow x_{T-1} \rightarrow \cdots \rightarrow x_0$, but during training each sample is trained on only a single, or at most a few, randomly sampled steps: $x_t \rightarrow x_{t-1}$. Consequently, it is impossible to cover the complete generation trajectory for each sample. In contrast, in autoregressive training, next-token prediction with causal attention ensures that the supervision signal covers every step of the generation process for each sample.

These two challenges result in discrete diffusion models performing worse than autoregressive models **under the same training budget**. Empirical evidence is presented in Sec. 4.

## 3 METHODOLOGY

### 3.1 TRAINING

We adopt two strategies to mitigate the limitations of discrete diffusion training. The first applies autoregressive training prior to discrete diffusion training to enhance the model's learning of multimodal understanding ability. The second initializes the model with the weights of an autoregressive MLLM, followed by discrete diffusion training to enable the acquisition of parallel decoding capabilities. Details of the training configuration, masking strategies, and sample padding are provided in the supplementary material.

**Strategy I (Dimple): Starting from a Diffusion Large Language Model** We denote the model obtained using Strategy I as Dimple. We initialize the model with Qwen2.5-VL's vision encoder (Bai et al., 2025), DLM Dream (Ye et al., 2025b), and a randomly initialized two-layer projector (Liu et al., 2023a). Following the alignment and instruction-tuning paradigm of LLaVA (Liu et al., 2023b), we design a three-stage training pipeline: Autoregressive Alignment (AA), Autoregressive Instruction Tuning (AT), and Diffusion Instruction Tuning (DT).

In AA and AT, we modify Dream's attention mechanism by replacing the full attention mask with a causal attention mask and applying a next-token prediction loss. We adopt the LLaVA-NEXT (Liu et al., 2024) training recipe and its datasets. During AA, only the projector is trainable; during AT, all components except the vision encoder are trainable. In the experiments reported in Sec. 4, we refer to the model obtained after AA as the aligned model. In the DT phase, we transition back to the diffusion-based generation framework. The full attention mask in Dream is restored, and the loss defined in eq. (3) is applied. The LLaVA-NEXT instruction-following dataset is reused for this phase. All components except the vision encoder are trainable.

**Strategy II (Dimple+): Starting from a Autoregressive Multimodal Large Language Model** We denote the model obtained using Strategy II as Dimple+. For initialization, we adopt the full Qwen2.5-VL model (Bai et al., 2025). Training consists of a single Diffusion Instruction Tuning (DT) phase. Specifically, we modify the attention mask in Qwen2.5-VL to full attention and apply the loss defined in eq. (3), updating all components except the vision encoder. The training data consists of the instruction-following dataset from LLaVA-NEXT (Liu et al., 2024) and 1.3M purely text-based instruction-following samples collected from the MAmmoTH-VL (Guo et al., 2025) dataset. Since Qwen2.5-VL (Bai et al., 2025) has already undergone autoregressive alignment and autoregressive instruction tuning, this implementation actually delegates the AA and AT phases to Qwen2.5-VL.

### 3.2 INFERENCE TECHNIQUES

**Review: Confidence-Based Decoding.** In discrete diffusion models, the generation process proceeds iteratively over a fixed number of steps. At each step, a subset of masked positions is selected

and updated based on their confidence scores. Let $\mathbf{x}_t$ denote the input sequence at step $t$. Let $\mathbf{z}_t \in \mathbb{R}^{L \times V}$ be the corresponding logits, where $V$ is the vocabulary size. Taking MaskGIT's decoding algorithm (Chang et al., 2022) as an example, the decoding process is as follows.

Probabilities: $\mathbf{p}_t = \mathrm{softmax}(\mathbf{z}_t/\tau)$; Confidence: $c_t^{(i)} = \max(p_t^{(i)})$;

Select $K$ positions: $\mathcal{I}_t = \mathrm{TopK}(c_t, K)$; Sample tokens $x_t^{(i)} \sim \mathrm{Categorical}(p_t^{(i)})$ for $i \in \mathcal{I}_t$.

This approach ensures that tokens with higher prediction confidence are updated earlier.

**Confident Decoding.** In previous confidence-based decoding (Chang et al., 2022; Ye et al., 2025b; Nie et al., 2025), the number of tokens decoded per step is fixed. However, we argue that decoding should adapt to the semantic structure of the text: some steps may allow many tokens to be confidently predicted, while others may necessitate more caution. We therefore propose **Confident Decoding**, which dynamically adjusts the number of tokens updated per step based on a fixed confidence threshold $\gamma \in (0,1)$. Formally, at each step $t$, we define $\mathcal{I}_t = \{i \mid c_t^{(i)} \geq \gamma\}$, where $c_t^{(i)}$ is the confidence score for position $i$. If $\mathcal{I}_t$ is non-empty, all tokens at positions $\mathcal{I}_t$ are decoded. Otherwise, we decode the token at the position with the highest confidence. This method enables:

- decoding multiple tokens simultaneously when model is highly confident, improving efficiency;
- avoiding low-confidence updates, preserving generation quality.

This adaptive mechanism reduces unnecessary updates of uncertain tokens and enhances both quality and speed of generation.

**Prefilling.** Let $L_{\mathrm{prompt}}$ denote the lengths of the prompt, which includes the question, image, and system prompts. Let $L_{\mathrm{answer}}$ denote the length of the response. With vision tokens, the prompt can become unignorably long. The use of a full attention mask in diffusion decoding results in a quadratic complexity $\mathcal{O}((L_{\mathrm{prompt}} + L_{\mathrm{answer}})^2)$ per decoding step. To alleviate this cost, we re-implement the **Prefilling** strategy in the autoregressive models, which saves the key-value pairs of the prompt tokens after the first generation step and reuse them in the following steps, reducing the complexity to $\mathcal{O}(L_{\mathrm{answer}}^2)$. But, due to the use of a full attention mask in dMLLM, the prefilling technique is not strictly lossless.

## 4 Experiments

All training configurations, evaluation metrics, generation hyperparameters used during evaluation, and other experimental details are provided in the supplementary material. The sensitivity analysis of the generation hyperparameters and additional ablation studies on the "AR-then-Diffusion" training strategy are also included in the supplementary material.

### 4.1 Benchmark Performance

**Comparison with AR Baseline.** We compare the performance of Dimple and Dimple+ with their corresponding AR baselines in Tab. 1. Each AR baseline is trained with the same data, parameter initialization, and total number of training iterations as Dimple and Dimple+, ensuring a fair comparison. As shown by the results, both Dimple and Dimple+ outperform their AR baselines. Dimple achieves an average score of 62.4%, 1.8% higher than its baseline, while Dimple+ surpasses the baseline on 9 of 12 benchmarks. These results indicate that, under the same training budget, dMLLM can match the performance of autoregressive models on instruction-following and visual understanding tasks. We also include the performance of LLaVA-NEXT (Liu et al., 2024) and Qwen2.5-VL (Bai et al., 2025) as references. Compared with Qwen2.5-VL, Dimple+ and its AR baseline improve on some benchmarks but regress on others. On the one hand, this results from insufficient task coverage in the training data, which improves performance on some benchmarks but degrades it on others. On the other hand, it also suggests that learning for parallel decoding is not entirely orthogonal to multimodal understanding; instead, the process of learning parallel decoding affects multimodal understanding ability.

> **Takeaways:** dMLLMs like Dimple can achieve performance comparable to their AR baselines, under the same training budget.

| Model | Dimple-7B (ours) | Dimple-7B (AR Baseline) | Dimple-7B+ (ours) | Dimple-7B+ (AR Baseline) | LLaVA-NEXT-7B | Qwen2.5-VL-7B |
|---|---|---|---|---|---|---|
| #Training Samples/Tokens | 1.3M/0.8B | 1.3M/0.8B | 2.6M/1.6B | 2.6M/1.6B | 1.3M/- | -/2.6T |
| Base LLM/MLLM | Dream(Qwen2.5) | Qwen2.5 | Qwen2.5 VL | Qwen2.5 VL | Vicuna-1.5 | Qwen2.5 |
| GQA | 59.24 | 59.1 | 63.2 | **64.1** | 64.8 | 60.0 |
| MMBench_en_test | 74.6 | 73.9 | **81.5** | 81.2 | 68.7 | 83.5 |
| MME_percpetion | 1514 | 1477 | **1558** | 1535 | 1519 | 1533 |
| MME_cognition | 432 | 457 | **627** | 610 | 332 | 620 |
| MME_total | 1946 | 1934 | **2185** | 2145 | 1851 | 2153 |
| POPE | 86.2 | 86.3 | **87.7** | 86.9 | 86.7 | 87.9 |
| MMMU_val | 45.2 | 45 | **50.2** | 49.1 | 35.8 | 58.6 |
| SQA_img | 77.1 | 77.2 | 79.1 | **83.6** | 72.8 | 52.1 |
| AI2D | 74.4 | 73.6 | **82.4** | 80 | 65.4 | 83.9 |
| ChartQA | 63.4 | 58.5 | 74.7 | **74.8** | 54.9 | 87.3 |
| TextQA_eval | 61.6 | 53.6 | **65.4** | 64.1 | 64.8 | 77.8 |
| OCRBench | 565 | 486 | **699** | 674 | 490 | 783 |
| MathVista_test_mini | 42.3 | 42.6 | **61.0** | 59.0 | 33.0 | 68.2 |
| MMVet | 41.2 | 37.8 | 47.5 | **49.2** | 47.3 | 67.1 |
| Average | 62.4 | 60.6 | **70.7** | 70.2 | 58.5 | 73.8 |

Table 1: Benchmark Performance Comparison. Dimple outperforms autoregressive baselines trained on similar data scales. In the table, the highlighted values in each row indicate the best-performing results among the following four models: Dimple, Dimple (AR Baseline), Dimple+, and Dimple+ (AR Baseline).

| Model | MME_p | MME_c | MME total | MMMU | AI2D | ChartQA | TextQA | MathVista |
|---|---|---|---|---|---|---|---|---|
| Dimple+ (ours) | **1558** | **627** | **2185** | **58.2** | **82.4** | 76.7 | **65.4** | **61.0** |
| LaViDA-L | 1366 | 341 | 1707 | 43.3 | 70.0 | 64.6 | 56.3 | 44.8 |
| LaViDA-D | 1463 | 389 | 1852 | 42.6 | 59.0 | 61.0 | 57.1 | 42.1 |
| LLaDA-V | 1507 | 491 | 1998 | 48.6 | 77.8 | **78.3** | 65.3 | 59.7 |

Table 2: Performance comparison with other Discrete Diffusion MLLMs. Our Dimple+ outperforms other models with a clear Margin on most of the benchmarks. In the table, MME_p, MME_c, and MathVista are abbreviations for MME_perception, MME_coginition, and MathVista_test_mini, respectively.

**Comparison with Other dMLLMs.** We compared the performance of Dimple+, LaViDA (Li et al., 2025a), and LLaDA-V (You et al., 2025) in Tab. 2. Dimple+ significantly outperforms the other two baselines on most benchmarks, achieving state-of-the-art results for dMLLM models. Moreover, training Dimple+ requires substantially lower computational cost. Specifically, Dimple+ is trained on only 2.6M samples, whereas training LLaDA-V involves more than 17M samples. This demonstrates the effectiveness of the "Autoregressive Model Initialization" strategy for dMLLMs.

> **Takeaways:** Initializing with a strong AR-MLLM can substantially enhance the performance of dMLLMs.

## 4.2 ABLATION ON PREFILLING

The following experiments investigate the impact of the prefilling technique on both performance and generation speed in dMLLMs. We conducted experiments with response lengths of 4 and 8, evaluating benchmark performance with and without prefilling. As shown in Tab. 3, prefilling does not lead to significant performance degradation on most datasets, with the average performance drop across all benchmarks being only 0.8%. This indicates that, in the current model, visual perception and the utilization of image tokens remain largely unchanged during text generation. Next, we examined the effect of prefilling on generation speed. By adjusting the batch size to 1 and 32, we simulated scenarios of low and high GPU utilization, respectively. The results show that: (1) under low GPU utilization (batch size = 1), prefilling yields an average speedup of 1.79×; (2) under high GPU utilization (batch size = 32), the acceleration effect is more pronounced, reaching up to 7×.

> **Takeaways:** Prefilling provides dMLLM with an almost lossless speedup on most benchmarks, up to 7×, by avoiding redundant computation of input key-value states.

| R.L. | Prefilling | Metric | B.S. | SQA | MMMU | MME | POPE | ChartQA | AI2D |
|---|---|---|---|---|---|---|---|---|---|
| 4 | × | ACC | 1 or 32 | 84.4 | 43.4 | 1946 | 85.0 | 51.7 | 74.5 |
| | | TPS | 1 | 34.8 | 21.3 | 22.7 | 22.4 | 21.5 | 22.9 |
| | | | 32 | 45.9 | 35.5 | 37.0 | 36.9 | 36.3 | 37.2 |
| | ✓ | ACC | 1 or 32 | 84.4 | 43.4 | 1822 | 85.1 | 51.3 | 74.4 |
| | | TPS | 1 | 49.3 (×1.32) | 19.9 (×1.67) | 21.3 (×1.63) | 20.7 (×1.65) | 20.1 (×1.69) | 21.5 (×1.62) |
| | | | 32 | 168.5 (×3.42) | 62.8 (×3.16) | 77.6 (×3.64) | 78.2 (×3.78) | 74.4 (×3.70) | 79.7 (×3.71) |
| 8 | × | ACC | 1 or 32 | 84.1 | 45.3 | 1900 | 84.8 | 58.4 | 74.5 |
| | | TPS | 1 | 29.2 | 19.3 | 19.3 | 19.5 | 19.1 | 19.7 |
| | | | 32 | 44.9 | 39.7 | 40.6 | 40.4 | 40.4 | 40.7 |
| | ✓ | ACC | 1 or 32 | 84.0 | 45.2 | 1905 | 84.0 | 58.4 | 74.2 |
| | | TPS | 1 | 49.2 (×1.54) | 18.1 (×2.06) | 21.4 (×2.10) | 20.7 (×2.07) | 20.0 (×2.12) | 21.6 (×2.07) |
| | | | 32 | 288.2 (×5.86) | 116.5 (×6.44) | 145.1 (×6.80) | 147.4 (×7.12) | 138.9 (×6.95) | 147.9 (×6.85) |

Table 3: Ablation Study on the Prefilling Technique. We compared model performance (ACC) and inference speed (Tokens Per Second, TPS) with and without the use of prefilling. Performance-related rows are highlighted in  blue . The underlined values indicate the speedup factor achieved with prefilling. "R.L." and "B.S." denote Response Length and Batch Size, respectively. The results demonstrate that although, in theory, prefilling is not lossless in dMLLMs, it causes minor drops in performance on most benchmarks while delivering substantial improvements in inference speed.

## 4.3 ABLATION ON CONFIDENT DECODING

We evaluate the Confident Decoding from two perspectives: *efficiency* and *performance*. Confident Decoding is designed to reduce the number of inference steps, thereby accelerating generation. The last row in Tab. 4 shows the actual number of decoding steps $k$ under a fixed response length of 8, along with the corresponding speedup factors (in parentheses). As shown, Confident Decoding reduces $k$ to 1.25 2.87, resulting in approximately 3× to 6× speedup.

| | *Steps* | CharQA | SQA | POPE | AI2D |
|---|---|---|---|---|---|
| MaskGit | 8 | 63.0 | 77.2 | 86.1 | 72.9 |
| | 2 | 52.2 | 76.2 | 85.5 | 71.1 |
| Entropy | 8 | 63.0 | 77.2 | 86.0 | 73.0 |
| | 2 | 54.9 | 76.5 | 85.1 | 72.9 |
| Random | 8 | 59.3 | 76.1 | 86.0 | 72.8 |
| | 2 | 57.1 | 76.0 | 85.2 | 71.0 |
| Confident | *Acc* | 62.5 | 77.1 | 86.4 | 73.1 |
| Deocding | *Steps* | 2.87 (x2.8) | 1.41 (x5.7) | 1.25 (x6.4) | 1.46 (x5.5) |

Table 4: Comparison of confident decoding with other decoding algorithms. Confident decoding significantly reduces the number of decoding steps while maintaining nearly the same performance, thereby improving decoding efficiency.

Next, we compare Confident Decoding with previous decoding strategies (Ye et al., 2025b; Nie et al., 2025): MaskGIT, Entropy, and Random. Unlike these methods, Confident Decoding is the first algorithm capable of dynamically adjusting the number of decoding steps, whereas the others require a predefined fixed value of $k$. For fairness, we set the number of decoding steps for the other strategies to either the response length (8) or one-quarter of it (2). As shown in Tab. 4, reducing the decoding steps to 2 leads to a substantial performance drop for other methods—up to 10% on CharQA for MaskGIT and Entropy. In contrast, Confident Decoding achieves performance comparable to the 8-step baselines with only about 2 steps. These results demonstrate that Confident Decoding enables faster inference without sacrificing accuracy.

> **Takeaways:** Confident Decoding effectively accelerates decoding by up to 6× through reducing the number of generation steps, while incurring only minimal performance loss.

## 5 GENERATION BEHAVIORS

In this section, we provide a showcase to demonstrate that dMLLMs exhibit fundamentally different decoding behaviors compared to autoregressive models. Unlike autoregressive models, discrete diffusion models offer the unique capability of **explicitly controlling the content** generated at arbitrary positions. This allows us to specify certain tokens as fixed in the final answer prior to generation. We refer to such tokens as **structure priors**.

**Question**: What is the common item in the two images?
**Response**: In the first image, there is a pair of black scissors. In the second image, there is a pair of black scissors and a pink plate with a pig face on it. The common item in the two images is the scissors.
**Response Length / #Remaining Tokens / Actual Iterations**:
64/42/30

**Generation History**:

| In | the | first | image | , | there | | is | a | pair | of | black | scissors | . | | In | the |
|----|-----|-------|-------|---|-------|---|----|---|------|----|-------|----------|---|---|----|----|
| - | - | - | - | - | - | - | 2 | 1 | 4 | 5 | 16 | 5 | 3 | 4 | - | - |

| second | image | , | there | | is | a | pair | of | black | scissors | and | a | pink | plate |
|--------|-------|---|-------|---|----|---|------|----|-------|----------|-----|---|------|-------|
| - | - | - | - | - | 2 | 2 | 13 | 12 | 15 | 14 | 17 | 18 | 19 | 23 |

| with | a | pig | 's | face | on | it | . | | The | common | item | in | the | two | images | | is |
|------|---|-----|-----|------|----|----|---|---|-----|--------|------|----|-----|-----|--------|---|----|
| 22 | 24 | 25 | 26 | 27 | 20 | 21 | 4 | 5 | - | - | - | - | - | - | - | - | - |

| the | scissors | . |
|-----|----------|---|
| 11 | 10 | 9 |

Table 5: Example: Structured Reasoning and Early Answering.

**Question**: Extract the information in the image.
**Response**: {"Date : "Apr 18, 2018", "Time: "12:00 AM",}
**Response Length/#Remaining Tokens/Actual Iterations**: 32/22/7

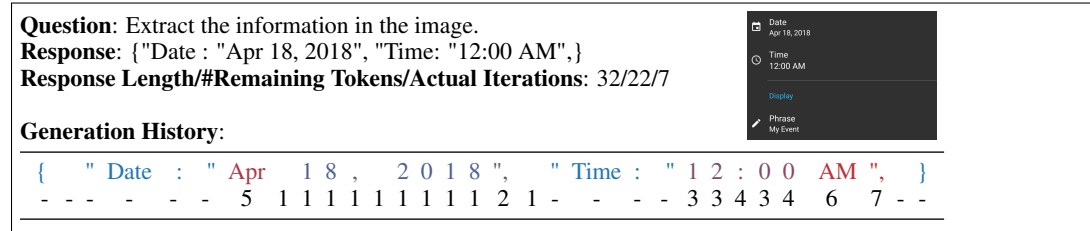

**Generation History**:

| { | | " | Date | : | " | Apr | | 1 | 8 | , | | 2 | 0 | 1 | 8 | ", | | " | Time | : | | " | 1 | 2 | : | 0 | 0 | | AM | ", | | } | |
|---|---|---|------|---|---|-----|---|---|---|---|---|---|---|---|---|----|---|---|------|---|---|---|---|---|---|---|---|---|----|----|---|---|---|
| - | - | - | - | | - | - | 5 | 1 | 1 | 1 | 1 | 1 | 1 | 1 | 1 | 1 | 2 | 1 | - | | - | | - | - | 3 | 3 | 4 | 3 | 4 | | 6 | | 7 | - | - |

Table 6: Example: Structured output and automatic semantic clustering during the generation process.

We present three illustrative examples. Each includes the input image, the question, and the generated response from Dimple. For each case, we annotate: the **response length** (i.e., the total number of tokens in the final answer), the **number of remaining tokens** that need to be generated after applying the structure prior, and the **actual number of decoding steps** (*actual iteration*) used in generation. Furthermore, we visualize the **generation history**, which records at which iteration each token was generated. Tokens provided by structure priors are marked with "-", and special tokens (e.g., padding) are excluded from the generation history but counted in the actual number of decoding steps. To enhance interpretability, we colored the response: tokens decoded earlier are shown in blue, while those generated later appear in red. Tokens with similar decoding steps share similar colors. All examples leverage the Prefilling technique and proposed Confident Decoding strategy.

## 5.1 EXAMPLE 1 (TAB. 5): STRUCTURED REASONING AND EARLY ANSWERING

In this example, the input consists of two images, and the model is asked to identify the common object present in both. We inject three structure priors into the expected answer: "In the first image, there ", "In the second image, there " and "The common item in the two images is". These priors guide the model to follow a reasoning trajectory resembling structured thought. Unlike the chain-of-thought (CoT) strategies used in autoregressive models, which rely on indirect guidance, structure prior enables direct and precise control over intermediate reasoning steps.

Remarkably, the correct answer "scissors" is decoded at the **10th** iteration, prior to the completion of the full response. The subsequent steps focus on completing the thinking trajectory. This illustrates a key strength of diffusion models: they can arrive at the final answer earlier in the generation process. In contrast, autoregressive models must complete the full sequence before producing the answer.

## 5.2 EXAMPLE 2 (TAB. 6): STRUCTURED OUTPUT

In this example, the task involves extracting visual information from an image. The question does not specify the desired output format and the attributes required. All of these are indicated by the

**Question**: How many objects are preferred by more than 90 percent of people in at least one category? Please think step by step.

Most preferred objects of different categories

Percent of People

**Response**: Nothing is preferred by more than 90 percent of people in any category. Thus, the answer is \box{Zero}.
**Response Length/#Remaining Tokens/Actual Iterations**:
32/23/16
**Generation History**:

| Nothing | is | preferred | by | more | than | | 9 | 0 | percent | of | people | in | any | category | . | Thus |
|---|---|---|---|---|---|---|---|---|---|---|---|---|---|---|---|---|
| 17 | 16 | 15 | 14 | 13 | 12 | 11 | 10 | 10 | 10 | 9 | 8 | 7 | 6 | 5 | 3 | - |

| , | the | answer | is | | box | { | Zero | }. |
|---|---|---|---|---|---|---|---|---|
| - | - | - | - | - | - | - | 4 | 2 |

**Response**: All of the objects in the chart have a maximum of 90 percent, but no object actually reaches 90. Therefore, none of the objects are preferred by more than 90 percent of people in at least one category. Thus, the answer is \box{Zero}.
**Response Length/#Remaining Tokens/Actual Iterations**: 64/55/37
**Generation History**:

| All | of | the | objects | in | the | chart | have | a | maximum | of | | 9 | 0 | percent | , | but |
|---|---|---|---|---|---|---|---|---|---|---|---|---|---|---|---|---|
| 20 | 17 | 15 | 16 | 18 | 19 | 38 | 21 | 22 | 26 | 23 | 25 | 28 | 24 | 27 | 29 | 33 |

| no | object | actually | reaches | | 9 | 0 | . | Therefore | , | none | of | the | objects | are | preferred |
|---|---|---|---|---|---|---|---|---|---|---|---|---|---|---|---|
| 34 | 35 | 36 | 37 | 31 | 32 | 30 | 13 | 14 | 12 | 11 | 11 | 10 | 8 | 7 | 6 |

| by | more | than | | 9 | 0 | percent | of | people | in | at | least | one | category | . | Thus | , |
|---|---|---|---|---|---|---|---|---|---|---|---|---|---|---|---|---|
| 6 | 6 | 6 | 6 | 6 | 6 | 6 | 6 | 5 | 5 | 5 | 5 | 4 | 5 | 9 | - | - |

| the | answer | is | | box | { | Zero | }. |
|---|---|---|---|---|---|---|---|
| - | - | - | - | - | - | 3 | 2 |

Table 7: Example: Length Control.

structure prior. In this example, the structure prior is "{date:", "time:" and "}". These priors define a JSON-like response layout, and the model successfully generates structured content accordingly.

This example also provides direct evidence of the effectiveness of **Confident Decoding**. The number of tokens decoded per iteration varies dynamically. For instance, 9 tokens corresponding to the date and year are decoded simultaneously in a single step. Despite a total of 22 tokens to be generated, the model completes generation in only 7 iterations, one-third of the total number of tokens. This parallelism significantly accelerates inference.

### 5.3 EXAMPLE 3 (TAB. 7): LENGTH CONTROL

Controlling output length is difficult in autoregressive generation. However, discrete diffusion models inherently support this capability. In this case, we specify structure priors at the end of the sequence. Specifically, we force tokens at positions $[-12 : -4]$ to be: "Thus, the answer is \box{". This allows us to strictly control where the model concludes its generation and places the final answer. We demonstrate two configurations with response lengths of 16 and 32. In each case, the model adjusts its reasoning span to fill the available token budget appropriately.

## 6 RELATED WORK

In the supplementary material, we also discuss the differences between Dimple and other approaches based on parallel decoding.

### 6.1 DIFFUSION LANGUAGE MODELS

The first problem when introducing diffusion process into language decoding, is the choice of a continuous diffusion space or discrete one. There are explorations in the continuous spaces. DiffuSeq (Yuan et al., 2023) use a partial noising and conditional denoising process to model sequence-to-sequence diffusion in continuous space. Argmax Flows and Multinomial Diffusion propose

continuous relaxations for categorical distributions for better generation stability (Hoogeboom et al., 2021). SED (Strudel et al., 2022) directly use the embedding space for the diffusion of natural language tokens. For the exploration in the discrete space, Structured Denoising Diffusion Models (SDDM) introduce discrete diffusion tailored to linguistic structures (Austin et al., 2023); RDM (Zheng et al., 2024) reparameterizes the backward discrete diffusion into a step sampling process to provide more flexible and unified framework. For training diffusion language model, loss function is the key. MDLM (Sahoo et al., 2024b), MD4 (Shi et al., 2025a) and SEDD (Lou et al., 2024) show that both a simple weighted mask language modeling loss and a score entropy loss, which is adopted from the score matching loss, are suitable for training. In this work, we follow the thread of masked language modeling scheme for the language diffusion process. In this direction, Diffusion-LM (Xu, 2022) applies diffusion to masked language modeling to enhance controllable text generation; DiffusionBERT (Sahoo et al., 2024a) uses pretrained BERT as an initialization of the diffusion training to accelerate convergence and improve performance. Besides using a pure diffusion language model, auto-regressive hybrids like AR-Diffusion (Li et al., 2023) and Semi-autoregressive methods like SSD-LM (Han et al., 2023) aim to combine the benefits of diffusion and autoregressive modeling. SSD-LM also shows that with a simplex projection the diffusion process can also be conducted in the natural vocabulary space. Recently, diffusion language models have been scaled to show comparable performance with the SOTA auto-regressive model, to learn instruction following ability and to have even better planning ability (Nie et al., 2025; Gong et al., 2025; Ye et al., 2025a;b).

## 6.2 MULTIMODAL LARGE LANGUAGE MODELS

Multimodal large language models (MLLMs) have rapidly progressed with several open-source series such as LLaVA, Eagle, InternVL, and Qwen-VL. LLaVA aligns vision encoders with LLMs via visual instruction tuning and demonstrates competitive performance using efficient training and external tool use (Liu et al., 2023b;a; Li et al., 2024; Liu et al., 2024). Eagle models explore vision-language integration through multiple encoders and long-context strategies, notably Automatic Degrade Sampling and Image Area Preservation (Azadani et al., 2025; Li et al., 2025b; Chen et al., 2025). InternVL focuses on scaling vision-language models, evolving from large vision encoders to joint multimodal pretraining, and achieves state-of-the-art results with models like InternVL2.5 and InternVL3 (Chen et al., 2024b;a; Mohbat & Zaki, 2024; Luo et al., 2025). Qwen-VL introduces multilingual vision-language models with dynamic resolution processing and bounding-box generation, achieving competitive performance across diverse tasks (Bai et al., 2023; Zhang et al., 2024; Bai et al., 2025).

With the advancement of discrete large language models (dLLMs), discrete diffusion has also been applied to multimodal understanding tasks. Concurrent with Dimple, models such as UniDisc (Swerdlow et al., 2025), MMaDA (Yang et al., 2025), LLaDA-V (You et al., 2025), and LaViDA (Li et al., 2025a) have emerged. Among them, UniDiscc (Swerdlow et al., 2025) and MMaDA (Yang et al., 2025) are unified discrete diffusion models that jointly model multimodal and text generation within the discrete diffusion framework. Dimple, LLaDA-V (You et al., 2025), and LaViDA (Li et al., 2025a) are all multimodal understanding models. In terms of training, LLaDA-V (You et al., 2025) successfully scales the training dataset to 17 million samples. LaViDA (Li et al., 2025a) introduces complementary masking strategies for training data, while Dimple proposes a hybrid training method called "First-Autoregressive-then-Diffusion". For inference, both Dimple and LaViDA (Li et al., 2025a) demonstrate the effectiveness of pre-filling techniques in discrete diffusion models. Additionally, Dimple introduces the Confident Decoding algorithm, the first method capable of dynamically controlling the number of generation steps in a discrete diffusion model, thereby balancing generation speed and quality.

## 7 CONCLUSION

In this work, we introduce Discrete Diffusion Multimodal Large Language Models **Dimple** and **Dimple+**, both of which match the performance of their autoregressive baselines while offering unique advantages such as parallel decoding and structure-aware generation. Our experiments validate the feasibility and strengths of the dMLLM paradigm, opening up new directions for efficient and controllable multimodal generation.

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
