# OpenReview forum: "Dimple: Discrete Diffusion Multimodal Large Language Model with Parallel Decoding"
_ICLR.cc/2026/Conference — Submitted to ICLR 2026_

### Official Review · Reviewer_YGWg · 2025-10-22

**Soundness:** 3
**Presentation:** 3
**Contribution:** 4
**Rating:** 6
**Confidence:** 3

**Summary:**

This paper introduces Dimple and Dimple+, multimodal large language model with discrete diffusion decoding.

The motivation is to address the instability and inefficiency of pure diffusion models in multimodal tasks.
And the paper validates two distinct strategies for converting AR models into diffusion-based multimodal models. (dimple and dimple+)

It also proposes a decoding strategy called “confident decoding” to reduce inference steps without obvious performance degradation.

**Strengths:**

1. The paper validates two practical strategies for converting AR models into diffusion-based multimodal models, providing clear guidance for the community.
* Dimple: Starts from a diffusion-only LLM without multimodal capability. It first applies autoregressive training to align vision and language, then switches to diffusion-based instruction tuning.
* Dimple+: Starts from a well-pretrained AR multimodal model (Qwen2.5-VL) and directly modifies its attention mask and loss function to enable diffusion training.
2. It introduces Confident Decoding, a dynamic decoding method that improves inference efficiency without compromising output quality.
3. From the code in supplementary material, the model is integrated into the Transformers library and will be fully open-sourced, providing a reproducible and accessible reference for the community.
4. Through comparisons with AR baselines under identical training data, and with state-of-the-art diffusion models like LLaDA-V, Dimple+ demonstrates the effectiveness of diffusion training for multimodal tasks.

**Weaknesses:**

1. Although Dimple+ improves over its AR baseline, it still falls short compared to its initial AR models (Qwen2.5-VL) on several benchmarks. In particular, the performance gap on ChartQA is substantial, despite the training data includes similar samples (2 epoch results is much better than 1 epoch training), raising concerns about catastrophic forgetting and the scalability of diffusion models.

2. The paper lacks analysis on the optimal number of decoding steps (k). While Confident Decoding dynamically adjusts k, it remains unclear whether increasing k beyond 8 could further improve performance, especially in complex tasks.

3. There is no direct comparison between the inference speed of Dimple+ and standard AR models. Without this, it’s difficult to assess the practical efficiency gains of diffusion decoding.

4. Despite the promising speedups and results reported for Confident Decoding, empirical testing with the open-sourced Dimple model shows noticeable degradation in long-context text generation. For example, when max_tokens is set to 64 or higher, even with step counts of 128 or 256, the model often fails to produce correct EOS tokens and instead generates excessive hallucinations. A controlled comparison with Dream-7B or other AR baseline would help clarify whether Dimple suffers from diffusion-induced degradation in textual capabilities.

**Questions:**

Overall, I think this is a strong paper that makes a meaningful contribution to the development of multimodal diffusion language models. However, I have a few questions that I hope the authors can clarify:
1. **Conclusion and Fair Comparison Between Dimple and Dimple+**
The paper presents two training strategies—Dimple and Dimple+, but does not offer a direct comparison under the same training data. Could the authors provide results showing how the two models perform when trained on identical datasets? This would help clarify whether initializing from a pretrained AR multimodal model is consistently more effective than learning multimodal alignment from scratch.
2. **Impact of Training Scale and Potential Forgetting**
Dimple+ inherits strong multimodal capabilities from Qwen2.5-VL, while Dimple builds them from new data. Given that Dimple+ is trained on limited data, is there a risk that diffusion tuning may gradually overwrite or degrade the original model’s capabilities? Could the authors comment on whether the performance gap between Dimple and Dimple+ might narrow or reverse with larger-scale training?
3. **Trade-off Between Performance and Latency**
The supplementary material reports performance under a maximum decoding step (k=8), but does not explore how performance scales with larger k. Could the authors provide results showing whether increasing k beyond 8 leads to further improvements? Additionally, a comparison of inference latency between Dimple+ and standard AR models like Qwen2.5-VL would be valuable to understand the practical efficiency trade-offs.

---

### Official Review · Reviewer_3Bzn · 2025-10-24

**Soundness:** 2
**Presentation:** 2
**Contribution:** 2
**Rating:** 2
**Confidence:** 4

**Summary:**

This paper explore the discrete diffusion for MLLMs, and present two diffusion MLLMs, termed Dimple and Dimple+. Besides, the authors also explore the parallel decoding to accelerate the inference of MLLMs.

**Strengths:**

1. The authors explore the diffusion paradigm for MLLMs, and investigate this manner under the setting of pure LLMs and MLLMs.

2. A confident decoding is proposed to accelerate the inference of MLLMs, which can achieve 2x-6x speedups.

**Weaknesses:**

1.  The experimental designs and results are hard to support the arguments. For instance, compared to the default QWen2.5-VL-7B, Dimple+ encounters obvious performance drops on multiple benchmarks.  Moreover, the results of Dimple-7B-AR baseline is also questionable. Compared to LLaVA-Next which uses a much weaker LLM, Dimple-AR-baseline perform worse on multiple benchmarks. These results are quite unreasonable. If the authors want to proof the merits of diffusion, they can make comparisons to open-sourced AR-MLLMs under the same experimental settings.

Besides, some notations of Tab.1 are also incorrect. If Dimple-7B use QWen2.5-VL as the base MLLM, the training samples should not be 1.6B. It is 2.6T+1.6B.

2. The actual contribution of Confident Decoding is unknown. From the paper, it is hard to recognize how many improvements and novelty the proposed Confident Decoding has compared to previous parallel decoding works. And how confident decoding implemented is also unclear.

Besides, can the parallel decoding only be achieved via the diffusion paradigm, or it can also be done in AR-MLLMs? It is important to the readers to judge the significance of diffusion MLLMs compared to existing MLLM research.

3. Following the first question, the comparison of DIMPLE to existing AR-MLLMs is not sufficient. More experiments should be conducted to show the merits of diffusion modeling.

**Questions:**

Q1 Can the parallel decoding only be achieved via the diffusion paradigm, or it can also be done in AR-MLLMs? It is important to the readers to judge the significance of diffusion MLLMs compared to existing MLLM research.

---

### Official Review · Reviewer_6WKY · 2025-10-29

**Soundness:** 3
**Presentation:** 3
**Contribution:** 3
**Rating:** 4
**Confidence:** 3

**Summary:**

The paper introduces **Dimple and Dimple+**, two discrete diffusion multimodal LLMs (dMLLMs) that bring parallel decoding to vision-language models. Dimple follows a hybrid training pipeline, autoregressive (AR) alignment and instruction tuning, then discrete diffusion, while Dimple+ starts from an AR MLLM (Qwen2.5‑VL) and acquires parallel decoding via diffusion-only tuning. The authors propose Confident Decoding (adaptive multi-token updates per step), re-use Prefilling with bidirectional attention to cache prompt KV states, and demonstrate Structure Prior for fine‑grained output control. On 12 benchmarks, Dimple/Dimple+ generally match their AR baselines; Dimple+ attains state-of-the‑art results among dMLLMs versus LaViDA/LLaDA‑V with notably fewer training samples. Confident Decoding yields ~2x-6x fewer steps with minor loss, and Prefilling offers ~1.7x-7x speedups with small average drops. Qualitative cases show structured reasoning, JSON‑like formatting, and length control.

**Strengths:**

- Two pragmatic routes to dMLLMs: AR->diffusion (Dimple) and AR‑MLLM initialization->diffusion (Dimple+). Confident Decoding adaptively sets the number of updated tokens per step, unlike fixed‑K schedules in prior work. Structure Prior gives direct positional control, early answering and enforced formats, difficult for AR models.

- Competitive accuracy vs. matched AR baselines on 12 benchmarks and clear SOTA among dMLLMs with far fewer training samples than LLaDA‑V. Ablations quantify Prefilling speedups ($\sim$1.7x-7x) with small average performance drop ($\sim$0.8\%) and show Confident Decoding reduces steps by $\sim$3x-6x at near‑baseline accuracy.

- Training strategies, attention masks, and losses are explicit; examples on pp. 7-8 make the decoding behavior and controllability intuitive.

- Encourages a practical path to parallel, controllable multimodal generation; speedups are meaningful for long‑prompt VLM inference.

**Weaknesses:**

- While Dimple+ is SOTA within discrete diffusion MLLMs, it lags a strong AR MLLM (Qwen2.5‑VL‑7B) on several tasks (e.g., ChartQA 74.7 vs. 87.3; OCRBench 699 vs. 783; Table 1), making the overall value proposition partly about parity plus speed/controllability, not raw accuracy.

- The paper claims parity under the same budget, but AR vs. diffusion differ in token supervision density and FLOPs/step; matching only by iterations or tokens may not equate compute. A FLOPs‑normalized comparison would strengthen the claim.

- Results are limited to ~7B scale; it is unclear how the approach scales to larger models or image‑dense prompts with much longer contexts.

- Structure Prior is shown qualitatively (Tables 5–7) but lacks metrics such as exact‑format accuracy, early‑answer timing/quality trade‑offs, or robustness when priors conflict with content.

- Confident Decoding hinges on the confidence threshold $\gamma$; the main paper does not show sensitivity curves or cross‑task robustness (Table 4 reports one configuration).

**Questions:**

- How exactly is "same training budget" defined and matched (tokens, optimizer steps, wall‑clock, or FLOPs)? Please provide FLOPs‑level accounting for AR vs. diffusion phases and rerun a compute‑matched comparison.

- How sensitive is Confident Decoding to $\gamma$ across tasks and lengths? Could you report curves (accuracy vs. steps vs. $\gamma$) and per‑task optima?

- Have you tried larger backbones (e.g., ≥14B)? Do speedups and parity hold at scale, and does Prefilling remain near‑lossless with very long image sequences?

- Can you quantify Structure Prior with exact-match format rates (e.g., strict JSON/LaTeX) and measure the earliness of correct answers vs. final sequence length? Any failure modes when priors are partially wrong or adversarial?

- Beyond the listed vision–language tasks, how do Dimple/Dimple+ perform on purely textual reasoning/planning benchmarks where diffusion LMs have reportedly excelled?

- For Dimple, can you show matched‑compute comparisons of (i) diffusion‑only vs. AR->diffusion, and (ii) varying the lengths of AA/AT/DT phases?

---

### Official Review · Reviewer_h84d · 2025-11-01

**Soundness:** 3
**Presentation:** 2
**Contribution:** 2
**Rating:** 4
**Confidence:** 4

**Summary:**

This paper introduces Dimple and Dimple+, two Discrete Diffusion Multimodal Large Language Models (dMLLMs) that leverage discrete diffusion for parallel decoding in multimodal (vision-language) tasks. Dimple is initialized from a discrete diffusion LLM and learns multimodal understanding via a hybrid training paradigm (autoregressive pretraining, then discrete diffusion). Dimple+ is initialized from an autoregressive MLLM and acquires parallel decoding via pure discrete diffusion training. Both models achieve performance comparable to their autoregressive baselines, with Dimple+ establishing new state-of-the-art results among dMLLMs.

The authors propose several inference and training innovations:

1.Confident Decoding: Dynamically adjusts the number of tokens generated per iteration, accelerating decoding by 2×–6× with minor performance degradation.

2.Prefilling: Adapts a technique from AR models to dMLLMs, achieving nearly lossless speedups of 1.7×–7×.
Structure Prior: Enables fine-grained control over response format and reasoning structure, which is difficult in AR models.

3.Extensive experiments on standard multimodal benchmarks show that Dimple and Dimple+ match or surpass AR baselines and outperform other dMLLMs, with strong ablations and qualitative analyses.

**Strengths:**

The strengths of the paper are:
1. Originality: Proposes a hybrid training paradigm (AR pretraining + diffusion) for dMLLMs, which is empirically effective. Introduces confident decoding and structure prior, enabling dynamic parallel decoding and fine-grained output control—capabilities not available in AR models.
2. Clarity: Clear motivation, methodology, and results presentation.
3. Significance: The proposed techniques (confident decoding, structure prior) are likely to inspire further research in efficient and controllable generation.

**Weaknesses:**

The weaknesses of the paper are:
1. Novelty: The proposed ideas are engineering centric. The confident decoding with flexible steps is proposed in literature I believe. I don't weigh too high for the overall novelty of the paper, but is okay in some sense.
2. Ablation on Structure Prior: While qualitative examples are provided, a more systematic quantitative evaluation of the structure prior’s impact would strengthen the claims.
3. Lack of comparisons and experimental details.

**Questions:**

1. Will you release code, models, and scripts for reproducibility?
2. I don't see supplementary materials of experimental details for reproducibility.
3. In table1, are you going to compare the inference speed of Dimple+ to Qwen2.5L or its AR baselines? It is interesting to know how much the diffusion LLM is faster than AR model.
4. Comparing Dimple and Dimple+, it is hard to draw the conclusion that initialization from AR training is better based on the experiment. The alignment and Instruction tuning phase of Dimple is done with much less training data and compute than Qwen2.5 VL. I believe the authors have used the 1.3M/0.8B data for this while Qwen2.5b-VL used many more, is it?

---

### Meta-Review · Area_Chair_ExVV · 2026-01-07

**Summary:**

This paper proposes Dimple and Dimple+, two discrete diffusion–based multimodal large language models designed to enable efficient parallel decoding for multimodal tasks. The work explores practical training routes for dMLLMs and introduces several techniques, including Confident Decoding, Prefilling, and Structure Prior.

Reviewers raised the following main concerns:
- Novelty: The contributions were viewed as largely engineering-driven, with limited conceptual novelty over existing approaches.
- Experimental evaluation: Multiple reviewers questioned the clarity of comparisons with the AR models such as Qwen-2.5-VL, as well as the observed performance gaps relative to AR baselines especially in some domains.
- Clarity and analysis: Reviewers noted unclear definitions of training budgets and insufficient ablation or quantitative analysis of proposed components, particularly Structure Prior and Confident Decoding.

There is no rebuttal provided. Overall, the majority of reviews leaned toward rejection.

**Reviewer Concerns:**

No responses are provided by the rebuttal.

**Reviewer Scores:**

Given no responses are provided, I do not expect a score change.

---

### Decision · Program_Chairs · 2026-01-26

Reject